# Intervention Works Conducted to Ensure the Stability of a Slope: A Sustainability Study

**Mircea Raul Tudorica** [1,2,*] and **Corneliu Ioan Bob** [3]

1     Department of Civil Engineering, Faculty of Constructions, Cadaster and Architecture, University of Oradea, 410058 Oradea, Romania

2     Doctoral School of Engineering Sciences, "Polytechnic" University of Timisoara, 300006 Timisoara, Romania

3     Department of Civil Engineering and Installations, Faculty of Civil Engineering, "Polytechnic" University of Timisoara, 300233 Timisoara, Romania; corneliu.bob@gmail.com

\*     Correspondence: tudorica.mircea@gmail.com

**Abstract:** Challenges related to sustainability arise in all areas of human activity, but with a significant impact on the environment considering that the construction industry is held accountable for nearly one-third of the world's final energy consumption. The aim of this paper is to assess through the use of the Bob–Dencsak specific model a sustainable slope design taking into account environmental, economic, and safety variables. Thus, analysis was performed on four intervention works, two versions of reinforced concrete retaining walls and two versions of reinforced soil with a biaxial geogrid, which ensure the stability of a slope that serves as a base for an access road to an ecological landfill located in Alba County, Romania. The study's analysis points out that reinforced soil retaining walls are far more sustainable, providing the best sustainability indices, which is also supported by the impact of geogrids compared to reinforced concrete, thus resulting in the finding that reinforced concrete is less sustainable, achieving increases of up to 23% for embodied energy and 66% of $CO_2$ emissions in the atmosphere. Finally, the paper provides recommendations for future research on the sustainability assessment of slopes, with the intention of reducing environmental damage, while keeping costs to a minimum.

**Keywords:** sustainability; safety factor; embodied energy; GHG gas emissions; retaining wall; reinforced concrete; environmental protection; geogrids; soil improvement

## 1. Introduction

The contextual meaning of the frequently used definition of sustainable development appears to be "The development that meets the needs of the present without compromising the ability of future generations to meet their own needs" [1,2]. The notion of sustainability, encompassing environmental, economic, and social aspects, is a significant topic in many different industries. The construction industry, as one of the most important industries in both developed and developing countries, has had a significant impact on various aspects of the environment, economy, and society. Sustainable construction has been introduced as a technique to assess the different stages of construction in recent years, in terms of social, economic, and environmental dimensions, also referred to as the three bottom lines (TBL) [3]. Energy demand and consumption have increased rapidly in recent years as a result of humankind's ever-increasing needs in the economy, industries, and agriculture [4]. Currently, a substantial amount of energy and material consumption is attributable to the construction industry [5]. As a consequence, the European Union has developed an interest in construction and its energy efficiency through innovative approaches, concentrating on present and future trends and concerns. Europe's priorities for the upcoming years until 2050 will be the decarbonization of the building industry and climate neutrality. This shift is highly intriguing as well, and it appears that a true revolution has started in this direction [6–8]. In response to the expanding demand for both urban and rural

infrastructure development, more investigations are necessary to ensure the safety of numerous civil constructions, especially embankments and highways [9]. The life cycle assessment (LCA) method is currently used worldwide by many nations and organizations to assess and examine the energy use and environmental impact of diverse projects [10]. This approach was initially focused primarily on the fields of building construction, and relevant practices and research on roads and railways, as well as the layers beneath them, have only started in the last ten years [11,12]. Using life cycle assessment (LCA), Chang et al. investigated the distribution of carbon emissions in a segment of the California high-speed railway [13]. Improving the life cycle assessment approach by including the cost, Chan A. made a comparative analysis of three types of road surfaces: new, reconstruction, and regeneration, where he found that while a cement road surface has the highest greenhouse gas emissions, it consumes less energy than an asphalt road surface [14].

Because geotechnical engineering acts as an integrator, bringing together different civil engineering sub-disciplines, its significance has grown over time. In the field of geotechnical engineering, landslides are a major problem that can cause catastrophic consequences like infrastructure damage and potential casualties. Apart from these, there are also those which have suffered severe functional and structural damage as a result of earthquakes, phenomenon that has become increasingly common in recent years around the world [15]. Any mitigation plan for this issue, which impedes development efforts, must start with an assessment of these geologic risks [16]. Masses of rocks, soil material, or muddy flows that slide down a slope due to modifications in the slope's natural stability are known as landslides [17]. The slope stability behavior is influenced both by internal factors, such as the physical–mechanical properties of the material like friction angle and cohesion, as well as by external factors, where we can include the amount of rainfall and seismic activity. In this direction, Saptono and Rezky conducted a case study in Southeast Sulawesi, Indonesia, to analyze the sensitivity of embankment slopes, using the coefficient of variation (CV) approach, mainly following the most important physical parameters, namely the internal friction angle and cohesion. The study's findings provide data showing that the internal friction angle has the greatest impact on the stability of embankment slopes and highlights the fact that for the highest value of the variation coefficient CV, there is a serious risk of producing an avalanche [18]. Another relevant case study for geotechnical engineering was conducted in Algeria, where Boubazine et al. investigated the occurrence of landslides in the Tarzoust region, based on geophysical approaches. Using Vertical Electric Soundings (VES) and the Seismic Refraction Method (SRM) for underground exploration, as well as Electrical Resistivity Tomography (ERT), it was proven that the combination of geological and geotechnical data with geophysical deterministic methods can help engineers and decision-makers in land management. In addition, this approach offers recommendations consisting of topographical, inclinometric, and piezometric monitoring to track landslides and the effectiveness of soil reinforcement measures [19]. In areas that are impacted by this phenomenon, preventing social, economic, and sustainable vulnerabilities requires an efficient and secure slope stabilization execution. Therefore, due to the complexity of the slope stability analysis, but also due to the lack of research in this field, more thorough analyses are required to measure several different parameters such as the ground's volumetric weight, the elasticity modulus of the earth's geological layers, or the slope's geometrical configuration, together with the addition of drains, vertical columns, retaining walls, and reinforcements [20].

The importance of the landslide-related effects of climate change was highlighted in this context by policymakers, scientists, designers, and engineers. Our society has new opportunities for dealing with the global energy crisis through the sustainable design of a large-scale civil engineering project like slope stability and landslide management. These resources provide a practical response to the environmental problems and the world's energy requirements. Even though slope stability is essential for maintaining public safety and protecting the infrastructure, it frequently has disastrous results, underlining the importance of creating long-lasting and efficient methods to reduce the risks related to

landslides [21]. In the meantime, the environment is unintentionally destroyed by construction activities, leading to the formation of numerous engineering slopes [22]. In addition to causing landslides and other natural disasters, these can also have an impact on the effectiveness and safety of constructions. Furthermore, as a result of the digging operation, a significant amount of soil subdivisions could migrate to the topsoil, reducing biodiversity, upsetting the ecological balance, and negatively impacting the long-term development of the local economy [23–25]. In this manner, Shen et al. describe the technologies that are frequently used in China, which combine soil improvement with bioremediation procedures. Even if the ecological restoration process is now highly mechanized, there are various problems that need to be studied further such as ecological restoration plans which are not designed with the local geographic conditions, the assessment of the ecological restoration sometimes being unclear due to a lack of quantitative data, restored slopes not being adequately monitored over the long term, and their environmental protection being ignored occasionally in the construction sector in an effort to increase profits. The authors also include a summary of the advantages and an assessment of their social impact [26]. Lastly, environmental restoration can help to improve the ecosystem and biodiversity's ability to function throughout addition to reducing landslides, soil erosion, and other local geological natural disasters [27,28]. A frequent situation is represented by the existence of expansive soils with a high concentration of hydrophilic mineral parts, like illite and montmorillonite, which are extremely sensitive to changes in water content. Their volume also fluctuates as a result of the variation in water quantity [29,30]. Since the expansive soil slopes are constantly expanding and contracting due to the action of the wet–dry cycles, cracks will develop on their surface. The resilience of slope soil will be reduced as more rainfall filters in, resulting in the development of shallow slope collapses [31,32]. Maintaining the long-term equilibrium of an expansive soil requires a slope support system that is highly sustainable [33]. To increase the flexible support structure's capacity to withstand, recover after, and react to the collapse of an expansive soil slope, systematic analysis and investigations are required to determine its structural stability in the future. Zhang et al. concluded that according to the life cycle evaluation analysis, the flexible support system uses approximately 50% less resources and energy and emits 10 times less carbon than the rigid support system based on the results of their research in this area [34]. Frischknecht et al. conducted an environmental assessment of the two types of retaining walls and compared the principles of the reinforced concrete retaining walls and those strengthened using geosynthetic materials. The analysis was performed on a slope of 3 m high and 1 m wide, which revealed that the environmental effect of the slope could be decreased by using geosynthetic-reinforced retaining [35].

In the specialized literature specific to the field of civil engineering, there are numerous standards dealing with structural safety, as well as various studies and models for assessing their sustainability, most of which are specific to new constructions and are developed during the design phase. Achieving the desired level of ensuring a sustainable environment on a global scale requires taking the best decisions to protect the environment through the rational and productive use of economic resources, all while meeting society's current needs without affecting future generations who will benefit from them directly. In this sense, the researchers' attention must be directed to the old existing structures, which may or may not present some structural, aesthetic, or energetic vulnerabilities due to the age of the materials and equipment used, in order to meet the needs of the present in terms of their safety and exploitation. However, special attention needs to be given to the land under the structure in question, both in the case of new or old structures, known in geotechnical engineering as the foundation soil. This natural resource is indispensable for both structures, being the most ancient building material, to which the choice of its resistance and stability characteristics is not an option, only their improvement through various mechanical or chemical technological processes but which inevitably result in higher costs. In the last decade, with the understanding worldwide that it is vital in all fields of activity to find and apply effective solutions to reduce emissions, the concept has

been extended to the total elimination of embodied energy consumption and greenhouse gas emissions into the atmosphere resulting from the consumption of building materials across all industries, starting with tracking the manufacturing process, transportation, and the equipment used for putting them into operation, and real interest has started to appear for the sustainability study of the soil layers in the construction field for railways, tunnels, dams, roads, and highways. Nevertheless, there is still a big gap in the specialized literature that directly targets the sustainability of soil foundations and the possible intervention works that must be conducted on them.

This paper carries out a sustainability investigation focused on four intervention works to ensure the stability of a slope. The analysis was carried out applying the Bob–Dencsak specific model, which presents a series of advantages such as the method's focus on all three factors associated with sustainability, having a wide range of applications and consisting only of quantifiable parameters. The main purpose is to compare different solutions, in order to determine which is most efficient from a sustainable perspective. Thus, two intervention works have been explored which involved soil reinforcement with a geogrid in the configuration of the slope 2:3 and 1:1, where the sustainability index was obtained as $SI_1 = 0.920$, for the first mentioned case, and $SI_3 = 0.951$ for the second case, and another two intervention works of reinforced concrete: a retaining wall with a height of 2.50 m situated at the base of the slope, for which a sustainability index $SI_2 = 0.779$ was obtained; and the second retaining wall of 6.40 m in height, which shows the lowest value of the sustainability index at $SI_4 = 0.573$. Following the final values in the slope sustainability analysis, we can assert that the reinforced soil retaining walls obtained the highest sustainability scores, being much more sustainable than the ones using reinforced concrete. This can be highlighted by comparing the reinforced soil configuration with a slope of 2:3 and that of the 2.5 m reinforced concrete retaining wall, where approximately the same amount of filling material was used. As a consequence, it turned out that the reinforced concrete's embodied energy is only 2.69 times, while the GHG gas emissions are 7.40 times higher than those generated by geogrids, resulting in an 18% more sustainable solution than the version with a reinforced concrete retaining wall.

## 2. Materials and Methods

### 2.1. The Assessement of the Slope Safety Factor

The safety factor $F_S$ of a slope is defined as the "ratio between the actual soil's shear strength value and the lowest possible shear strength value needed to avoid failure" or the rate that must be decreased in soil shear strength to push a slope toward collapse [36].

$$F_S = \frac{\tau_f}{\tau} = \frac{\sigma \times tg\varphi + c}{\sigma \times tg\varphi_m + c_m} \tag{1}$$

where $F_S$ signifies the slope stability factor, $\tau_f$ signifies the ground's available shear strength, $\tau$ signifies the required or mobilized shear strength, $\sigma$ signifies the normal stress, $\varphi, c$ signifies the soil's shear properties, $\varphi_m, c_m$ signifies the mobilized shear characteristics, $\varphi$ signifies the soil particle's coefficient of frictions, and c signifies the soil's cohesiveness.

However, the development of methods focused on the stability of the slope surface has been restricted by the lack of information on soil shear strength characteristics and their relationships with other soil properties [37]. The soil's shear capacity is the highest level of shear stresses that soil can withstand without collapsing, and it is determined based on the characteristics of cohesiveness and the internal friction angle between soil particles. Even though most modern technologies are used worldwide, it is impossible to guarantee slope safety in every situation. Furthermore, design and sizing standards have been established. These align strength and effectiveness guarantees with "safety" margins regarded as "comfortable" by experts in the field. Methods based on the concept of ultimate equilibrium assume a known sliding surface (real or possible) and admit $F_s = 1$ over the entire sliding surface. These methods are not based on a mathematical foundation, and their biggest deficiency is that it presumes an incipient failure. The method is used even in

the case of stable slopes, with $F_s > 1$, which leads to situations that obviously do not reflect the reality in the field [38].

The limit equilibrium techniques that Fellenius introduced in 1927 have resulted in significant improvements, which presume that resistance follows Coulomb's formula along the sliding line, splits the sliding soil volume included within the circular arc into slices, and assesses its equilibrium by reducing the forces and moments to zero. Since then, comparable approaches have also been established, which covers Janbu in 1954, Bishop in 1955, Simplified Bishop in 1960, Morgenstern and Price in 1965, Spencer in 1967, Simplified Janbu in 1973, and Sarma in 1973 [39]. Slope failure is far more complicated than the limit equilibrium approach has been able to simulate. In reality, failure does not occur concurrently along a single distinct normal surface, but rather a localized failure gradually expands over a larger failure surface. With the exception of strictly structural slope failures, like those governed by a discontinuity in a delicate rock mass, internal deformation is a crucial factor in the progression of these failures [40].

Methods using finite elements, abbreviated as FEM, are crucial for resolving stress–strain issues, especially in situations involving the interaction of soil–structure and slope stability [41]. In FEM, the structure and performance of geotechnical materials is analyzed by an elastoplastic simulation based on the Mohr–Coulomb failure criteria. It makes sense, therefore, that its application in the context of civil works safety be taken into account. Nevertheless, the analysis is not direct due to the distinctive nature of slope issues under unsaturated conditions in which suction has a major impact, where extra care must be taken to accurately replicate certain details [42]. The basic procedures in FEM include the discretization process, choosing approximations for functions, equation derivation, collecting element properties to form universal equations, and primary quantity calculation (e.g., displacements) and secondary calculation (e.g., stresses). Discretization is the process of breaking down a continuous material into a system of comparable small individual components (also known as finite elements), where each element is examined and handled separately. Physical properties or constitutive characteristics are assigned to each element, and matrices for the assembly's rigidity are generated [43]. FEM is a numerical method for estimating limit value solutions for a variety of partial differential equations. Theoretically, it fulfills every prerequisite needed for a comprehensive resolution of a slope stability issue [44].

### 2.2. Sustainability Assessment Models

Establishing sustainability performance can be carried out using a variety of models. Among the most widely recognized that are always being developed are the Building Research Establishment Environmental Assessment Method (BREEAM) and the Leadership in Energy and Environmental Design (LEED) [45].

The evolving and re-scoping of an understanding of what constitutes sustainable construction is reflected in the progress and constant improvement of different performance rating systems [46,47]. Worldwide, several types of complex models for determining a building's sustainability are offered in the specialized literature. They include a range of parameters from multiple perspectives that impact the sustainability research. The total number of parameters for every dimension, the importance of each dimension's proportion in the final result, and how they are classified from a sustainability perspective are shown in Table 1 [48].

**Table 1.** Establishing classification from the perspective of sustainability in construction.

| Sustainability Model | Ecological Dimension | Economic Dimension | Social Dimension | Construction Classification |
|---|---|---|---|---|
| UK 1990 (59) BREEAM | (100%) 59 | - | - | Insufficient, points <30 Good enough, points 30–85 Very good, points >85 |
| USA 1993 (57) LEED | (100%) 57 | - | - | Bronze, points 40–49 Silver, points 50–59 Gold, points 60–79 Platinum, points >85 |
| Japan 2001 (80) CASBEE | (70%) 56 | - | (30%) 24 | C Class, grades <0.5 B- Class, grades 0.5–1 B+ Class, grades 1–1.5 A Class, grades 1.5–3 S Class, grades >3 |
| International 1996 (14–122) SBTool Model | 48% | 24% | 24% | Acceptable, score <1 Good, score 1–3 Excellent, score >3 |
| CEN TC350 (51) | (33.3%) 16 | (33.3%) 17 | (33.3%) 18 | A score of 100 points is the maximum. The classification based on the obtained score |
| Romania 2010 (45) Bob–Dencsak | (40%) 21 | (30%) 11 | (30%) 13 | Very good, points >80 (>4) Good, points 60–80 (3–4) Acceptable, points 40–60 (2–3) Insufficient, points <40 (<2) |

In many cases, these global models indicate certain disadvantages:

- The models do not take into account all three aspects of sustainability;
- They have a large number of parameters, some of which are difficult or impossible to quantify;
- The instruments are primarily designed for complete buildings, though they can be used, albeit with some difficulty for other kinds of construction projects and tasks.

To avoid the previously mentioned disadvantages which characterize global sustainability models, some specific models were proposed and applied, with the purpose of serving engineers in assessing the sustainability of certain particular construction works. The most significant advantages of these specific models are:

- They deal with all three aspects of sustainability;
- A wide range of application;
- They consist only of quantifiable parameters.

The main purpose of the specific models is to compare different solutions, in order to determine which is the most efficient from a sustainable point of view. Thus, a relative value is obtained for each solution, which is compared to the ideal value [49]. A similar approach has been proposed by Ding [50] and Diaz–Balteiro and Romero [51], but there are certain difficulties with using the models.

*2.3. Bob-Dencsak Specific Sustainability Model*

Based on fundamental mathematical formulas, the specific model logically combines the results of the parameters that were quantified to achieve a sustainability index SI.

$$\text{SI} = S_{\text{env}} + S_{\text{eco}} + S_{\text{soc}} \qquad (2)$$

$$S_{\text{env}} = \sum_{i=1}^{n} \alpha_i \times \frac{P_{i,\text{env}}^{R}}{P_{i,\text{env}}} \qquad (3)$$

$$S_{eco} = \sum_{i=1}^{n} \beta_i \times \frac{P_{i,eco}^{R}}{P_{i,eco}} \tag{4}$$

$$S_{soc} = \sum_{i=1}^{n} \gamma_i \times \frac{P_{i,soc}^{R}}{P_{i,soc}} \tag{5}$$

where: SI represents the sustainability index, $S_{env}$, $S_{eco}$, $S_{soc}$ represent the sustainability indexes to the social, economic, and environmental aspects, $\alpha_i$, $\beta_i$, $\gamma_i$ represent how each parameter in the environmental, economic, and social dimensions is rated, $P_{i,env}^{R}$, $P_{i,eco}^{R}$, $P_{i,soc}^{R}$ represent the reference value for each parameter, and $P_{i,env}$, $P_{i,eco}$, $P_{i,soc}$ represent the calculated values for each parameter.

If two or more solutions are compared, the values obtained as references can be regarded as the optimal values from each parameter; when conducting a self-assessment, the best practices that are currently available are used as standards. For those circumstances, where a parameter's higher value is thought to be more sustainable, in Equations (3)–(5) the parameters in the ratio of the reference value and the calculated one will be reversed. The final result of the developed specific model is the sustainability index SI, with a dimensionless value between 0 and 1, in which 0 represents the worst value and 1 the best value [52].

## 3. Case Study

### 3.1. A Brief Description of the Geographic Location

The examined objective is situated to the north-east of the municipality of Alba Iulia, at about 20 km in the north of Galda de Jos Village. The area in question has a polygonal shape, with a total surface of approx. 22 ha out of which 7 ha will be occupied by Cell 1. The ground surface records level differences from 277 m to 320 m, with a gentle slope of 1:8 . . . 1:10 in the direction southwest–northeast. Before the works started, the investigated settlement in its natural form did not show instability phenomena affecting the analyzed perimeter or the slopes from the settlement vicinity. The settlement is crossed from SW to NE, respectively, on the lines of the greatest slopes by ravines with depths of 0.5 to 2 m and slopes of 1:1 . . . 1:1.5 with a high potential for losing local stability. The geotechnical study involved performing 10 bore-holes within the site, as shown in Figure 1.

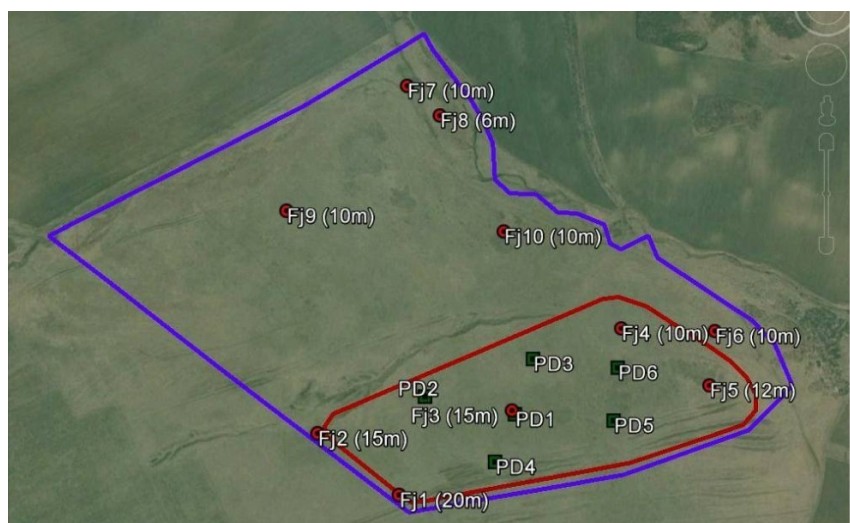

**Figure 1.** Bore-holes' arrangement within the site.

The geotechnical investigation revealed that the foundation soil is composed of a package of cohesive materials like clay—sandy loam, brown-yellow color, in a state of consistency from plastic to hard, located beneath a layer of vegetable soil varying in thickness from 10 to 30 cm.

Slope stability analysis was performed with the Geostru Slope application, with multiple tactics concerning the shape of failure surfaces, using the circular surface (the simplest shape). To avoid the situation of ultimate equilibrium, an acceptable safety level of 1.50 was proposed. The step search was set to 30, with a number of 30 strips in order to have a reasonable time period for the stability analysis. The partial coefficients for soil geotechnical parameters were considered at 1.25 for the angle tangent of internal friction and for the effective cohesion, respectively, 1.40 for the undrained cohesion.

The characteristics of the soil foundation and the adjacent soil layers were both introduced in the program according to the analyzed transverse profile, located next to the borehole Fj6, as can be seen in Table 2 [53].

**Table 2.** Characteristics of the slope analyzed.

| Layer No. | c (kN/m$^2$) | $\varnothing$ (deg) | G (kN/m$^3$) | G$_S$ (kN/m$^3$) |
|:---:|:---:|:---:|:---:|:---:|
| 1 | 33.5 | 11.62 | 18.63 | 20.59 |
| 2 | 0 | 30 | 19.93 | 21.13 |
| 3 | 56.60 | 15.30 | 20.53 | 23.76 |

where c represents the cohesion, $\varnothing$ represents the friction angle, G represents the specific weight, and G$_S$ represents the saturated specific weight.

### 3.2. The Intervention Works Analysed on the Slope

The stability factor's analysis was conducted using the computational application with imposed surfaces in Geostru Slope, which is based on the Finite Element Method (FEM).

The geometric configurations of the intervention works analyzed (two retaining walls made of reinforced concrete and two soil reinforcement with a biaxial geogrid) in the sustainability study are presented in Figure 2.

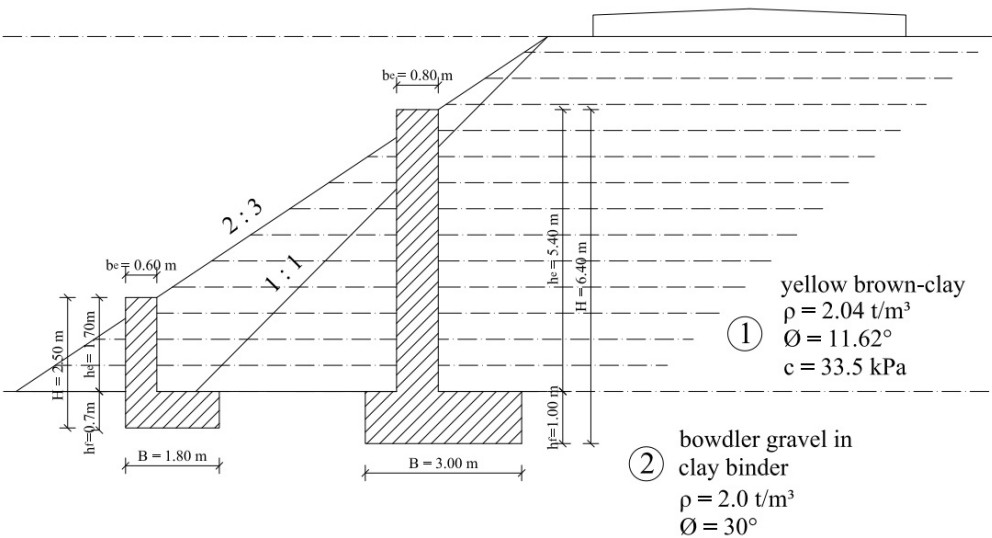

**Figure 2.** Geometrical configuration of the slope.

In addition to the intervention works that are the focus of this sustainability study, the initial research contained two additional unreinforced slope configurations, one with a slope of 2:3 and the other with a 2:3 slope to which a 2 m-wide berm was inserted halfway up the retaining wall's height, but which do not correspond in terms of resistance and stability, obtaining a safety factor below S$_F$ < 1.50, which represents the minimum accepted value. The safety factor's assessment is not the subject of this article; it is very well detailed in [53]. To evaluate the sustainability indices, we will analyze the following intervention works:

1. Reinforced soil, inclination of slope 2:3;

2. Retaining wall H = 2.50 m;
3. Reinforced soil, inclination of slope 1:1;
4. Retaining wall H = 6.40 m.

The technical data and the cost of the materials used in the intervention works, as well as the coefficients regarding the embodied energy and gas emissions in terms of manufacturing, transport, and equipment or machines used in the construction process, are given in Table 3.

**Table 3.** Materials, technical information, and coefficients utilized in the sustainability study.

| Material Used in the Intervention Works | Material Density (kg/m$^3$) | Embodied Energy Coefficient EE (MJ/kg) | GHG Emissions Coefficient EC-CO$_2$ (kgCO$_2$/kg) | Material Cost (EURO/m$^3$) |
|---|---|---|---|---|
| Stabilized filler material | 2240 | 0.082 | 0.0048 | 15 |
| Concrete C20/25 | 2400 | 0.91 | 0.131 | 100 |
| Steel bar $\varnothing$12 and $\varnothing$14 | 7850 | 29.20 | 2.59 | 9812.5 |
| Biaxial geogrid polypropylene | 900 | 101.30 | 4.10 | 10,385 |

The quantities of materials required for performing the intervention works were estimated for a section of 5 m in length, following the profile with the highest level difference, measuring nearly eight meters.

Thus, the analysis started with the retaining wall made of reinforced soil with geogrilles in the geometric configuration with the smoothest slope of 2:3, where it is necessary to bring and compact 424 m$^3$ of local soil. Biaxial geogrids with a specific weight of 0.284 kg/m$^2$ were used in both reinforced soil configurations, being arranged in layers along the entire height of the retaining wall, at distances of 50 cm between them, consolidating each layer on a 50 m$^2$ surface. The required quantity was established taking into account the design length of 10 m, which was supplemented with the overlapped length required to secure the geogrid to the layer above over a distance of one meter.

The reinforced concrete retaining walls were designed with the C20/25 concrete class and an S500 reinforcement mark. The retaining wall with a height of H = 2.50 m, located at the base of the slope, is characterized by an elevation width $b_e$ = 60 cm, foundation height $h_f$ = 70 cm, and foundation width B = 1.80 m. The retaining wall's elevation and foundation are reinforced with $\varnothing$12/10 cm bars on both directions, forming closed edges both in the longitudinal and cross sections, obtaining 745.75 kg of iron for the analyzed intervention work.

In the last intervention work, we have a retaining wall with a height of H = 6.40 m located very close to the ecological landfill's access road, which has the following geometric dimensions: elevation width $b_e$ = 80 cm, foundation height $h_f$ = 1.00 m, and foundation width B = 3.00 m. And in this case, closed edges are formed on both sections from bars $\varnothing$14/10, obtaining a total steel amount of 2512 kg.

With the exception of the filler material that was brought to the site from a distance of maximum 10 km away, all materials were delivered to the site from the nearest local construction supplies warehouse, positioned 30 km away. All materials were delivered in trucks that could transport between 3.5 and 20 tons, which have the following coefficients: embodied energy EE = 4.60 MJ/tkm and GHG gas emissions EC-CO$_{2e}$ = 0.28 kgCO$_2$/tkm.

The initial embodied energy, which depends on the embodied energy (EE) in terms of manufacturing materials, transport, and equipment or machines used in the construction process (En), is calculated with Formula (6).

$$\text{En} = \text{EE} \times \text{m} \tag{6}$$

The GHG gas emissions which resulted from construction materials in terms of manufacturing materials, transport, and equipment or machines used in the construction process (G) are calculated with Formula (7).

$$\text{G} = \text{CO}_{2eq} \times \text{m} \tag{7}$$

## 4. Results and Discussions

The results of the sustainability study were obtained using the Bob–Dencsak specific model, thus calculating all the parameters in question. The ecological dimension is represented by the consumption of embodied energies (En) and the total amount GHG gas emissions (G) in the process of manufacturing and transporting the materials used in the intervention works. In the results of the calculation, these factors are given equal weight, accounting together for 40% of the sustainability indices' value. The economic dimension of sustainability is expressed through the labor (W) and material costs (C) required to complete these interventions works, which also represent 40% of the final result, divided equally among the parameters within the dimension. The safety factor ($S_F$) expresses the social dimension of sustainability, assigning 20% of the final value of each intervention work within the sustainability study.

The quantities of materials used for each intervention work, as well as the data regarding the environmental dimension through the embodied energies and gas emissions from each material, are presented in Table 4.

**Table 4.** The quantities of materials used and the embodied energy and GHG emissions for each intervention work.

| Intervention Work | Name of Material | Quantity Volume (m$^3$) | Energy (MJ) | GHG Emission (kgCO$_2$) |
|---|---|---|---|---|
| 1. Reinforced soil, inclination of slope 2:3 | Filler Material | 424 | 121,569.28 | 7218.18 |
| | Biaxial Geogrid | 0.23 | 21,230.97 | 859.89 |
| 2. Retaining wall, H = 2.50 m | Filler Material | 412 | 118,128.64 | 7013.89 |
| | Concrete C20/25 | 11.70 | 29,427.84 | 3914.35 |
| | Steel Bar | 0.12 | 27,636.40 | 2447.70 |
| 3. Reinforced soil, inclination of slope 1:1 | Filler Material | 366 | 104,939.52 | 6230.79 |
| | Biaxial Geogrid | 0.23 | 21,230.97 | 859.89 |
| 4. Retaining wall, H = 6.40 m | Filler Material | 315 | 90,316.80 | 5362.56 |
| | Concrete C20/25 | 36.60 | 92,056.32 | 12,244.90 |
| | Steel Bar | 0.33 | 76,000.09 | 6731.16 |

Based on the previous quantities of materials used in the sustainability study, the economic dimension regarding the cost of materials and the cost of labor is presented in Table 5.

The final values of the coefficients which were used in the sustainability analysis of each individual intervention work are presented in Table 6.

**Table 5.** The cost of the materials and labor involved in the sustainability analysis.

| Intervention Work | Name of Material | Quantity Volume (m³) | Material Cost (EURO) | Labor (man × h) |
|---|---|---|---|---|
| 1. Reinforced soil, inclination of slope 2:3 | Filler Material | 424 | 6360 | 100 |
| | Biaxial Geogrid | 0.23 | 2389 | 128 |
| 2. Retaining wall, H = 2.50 m | Filler Material | 412 | 6180 | 97 |
| | Concrete C20/25 | 11.70 | 1170 | 16 |
| | Steel Bar | 0.12 | 1178 | 160 |
| 3. Reinforced soil, inclination of slope 1:1 | Filler Material | 366 | 5490 | 85 |
| | Biaxial Geogrid | 0.23 | 2389 | 128 |
| 4. Retaining wall, H = 6.40 m | Filler Material | 315 | 4725 | 75 |
| | Concrete C20/25 | 36.60 | 3660 | 50 |
| | Steel Bar | 0.33 | 3239 | 320 |

**Table 6.** The coefficients values used in the sustainability analysis.

| Intervention Work | Environmental | | Economic | | Social |
|---|---|---|---|---|---|
| | Energy (MJ) | GHG (kgCO₂) | Material (EURO) | Labor (man × h) | Safety Factor |
| 1. Reinforced soil, inclination of slope 2:3 | 142,800.25 | 8078.07 | 8749 | 228 | 2.18 |
| 2. Retaining wall, H = 2.50 m | 175,192.88 | 13,375.94 | 8528 | 273 | 2.05 |
| 3. Reinforced soil, inclination of slope 1:1 | 126,170.49 | 7090.68 | 7879 | 213 | 1.65 |
| 4. Retaining wall, H = 6.40 m | 258,373.21 | 24,338.62 | 11,624 | 445 | 2.01 |

The sustainability index for each intervention work is calculated with the formula:

$$\text{SI} = 0.2\frac{\text{En}^\text{R}}{\text{En}} + 0.2\frac{\text{G}^\text{R}}{\text{G}} + 0.2\frac{\text{C}^\text{R}}{\text{C}} + 0.2\frac{\text{W}^\text{R}}{\text{W}} 0.2\frac{\text{S}_\text{F}}{\text{S}_\text{F}^\text{R}} \tag{8}$$

where the reference values are: $\text{En}^\text{R} = 126170.49$ MJ, $\text{G}^\text{R} = 7090.68$ kgCO₂, $\text{C}^\text{R} = $ EUR 7879, $\text{W}^\text{R} = 213$ man × h, and $\text{S}_\text{F}^\text{R} = 2.18$.

Thus, after performing the calculations, the following sustainability index values were obtained: $\text{SI}_1 = 0.920$ for the reinforced soil, with a slope inclination of 2:3, $\text{SI}_2 = 0.779$ for the retaining wall with the height of 2.50 m, $\text{SI}_3 = 0.951$ for the reinforced soil, with a slope inclination of 1:1, and $\text{SI}_4 = 0.573$ for the retaining wall with the height of 6.40 m.

The initial objective of this project was to ensure the resistance and stability of a slope that serves as an access road to an ecological landfill, where the intervention works involve the use of local ground that is used as a filling material to support the road. Thus, the first intervention analyzed was a retaining wall with unreinforced soil in the most stable geometric configuration, with a slope of 2:3, where the safety factor was found to be less than the minimum acceptable by the current regulations. In order to obtain an acceptable safety factor, the base of the slope was increased by adding a berm of 2 m wide at the midpoint of the retaining wall's height, but also with an unfavorable result in terms of the resistance and stability of the slope, obtaining a value that is less than the 1.50 minimum

acceptable value, which is a mandatory requirement. Each of the analyzed variants fulfill the condition of resistance and stability, trying to achieve a high value for the safety factor, a fact that plays a major role in raising the social dimension.

In the version with the retaining wall reinforced with biaxial geogrids, with a slope of 2:3, a safety factor value of $F_S = 2.18$ was obtained, which was used as the social criterion's reference value when analyzing the sustainability of the intervention works. The 2.50 m-high reinforced concrete retaining wall placed at the base of the slope reduces its base by approximately 2 m, which leads to a greater storage capacity, but it strengthens the slope by increasing the base's stiffness, obtaining a satisfactory safety factor $F_S = 2.05$, which is comparable to the reinforced soil with a slope of 2:3. Due to the retaining wall's small dimensions, as well as the difference in filling material required to perform the intervention work, the total cost of the materials is lower than in the case of reinforced soil with a slope of 2:3, but taking into account all the parameters it offers a 18% lower sustainability index.

Investigating an additional decrease in the slope's base, which directly implies an overall reduction in the slope's stiffness due to its lack of massiveness, the geometric configuration of the reinforced soil retaining wall with a slope of 1:1 was analyzed, where a lower safety factor value $F_S = 1.65$ resulted. Although it has the lowest stability factor of all the configurations that were examined, this intervention work provides the reference values for the economic and ecological dimensions because it requires 58 m$^3$ less filling material than the reinforced soil retaining wall with a 2:3 slope configuration.

The geometric configuration of the slope with the smallest base, which provides an adequate total stiffness due to the large amount of concrete and steel bars used, is the intervention work of the reinforced concrete retaining wall with a height of 6.40 m, which offers a more than acceptable stability factor of $F_S = 2.01$. Due to the large volume of reinforced concrete, it recorded the highest values for the embodied energy En = 258373.21 MJ and GHG gas emissions G = 24338.62 kgCO$_2$, as well as the cost of materials and labor.

## 5. Conclusions

After carrying out the sustainability study of the four intervention works, it has been clearly observed that the best value SI$_3$ = 0.951 was obtained for the reinforced soil with a biaxial geogrid, with a slope of 1:1. Based on the parameters' values that were determined for this intervention work, it is important to draw attention to the optimal balance between the materials' energy consumption and gas emissions as well as their costs, including labor, as these represent the study's reference values. All of this supports the result, including the fact that the slope's geometrical configuration is stable, even if it recorded the lowest value of the safety factor, which is 10% higher than the minimum accepted value.

The second option is represented by the reinforced soil retaining wall with a slope of 2:3, with a sustainability index value of SI$_2$ = 0.920. Compared to the version with a slope of 1:1, there were increases in energy consumption of 13.5% and in the amount of gas released into the atmosphere of about 14% due to the filling material that must be brought additionally in order to achieve the slope geometrical configuration.

The third intervention work option from the perspective of sustainability, with an index value of SI$_3$ = 0.779, is represented by the reinforced concrete retaining wall with a height of 2.50 m placed at the slope base. Following the ecological dimension, if we make a comparison with the version of reinforced soil with a slope of 2:3, there are significant increases of 23% in energy consumption, respectively 66% for GHG gas emissions released into the atmosphere. This is clearly underlined by the material values in terms of embodied energy and GHG gas emissions, where for the geogrid the following values were obtained En = 21230.97 MJ and G = 859.88 kgCO$_2$ and respectively for reinforced concrete En = 57064.24 MJ and G = 6362.05 kgCO$_2$.

The lowest value of the sustainability index SI$_4$ = 0.573 was obtained by the reinforced concrete retaining wall with a height of 6.40 m. This result is highlighted by the values obtained by all of the studied parameters, specific ecological and economic dimensions, due to the large volume of reinforced concrete required, which unavoidably raises the total cost

of the intervention work. Even though the storage capacity of the ecological landfill was not a criterion, for further research it should be noted that this intervention work provides the smallest base of the slope which offers the biggest storage capacity. This aspect can also be taken into account in the case of intervention works to stabilize slopes that have a limited base for various reasons, such as the presence of railroads nearby, or any type of structure, or even just the simple existence of flowing water.

As a final conclusion, we can state that compared to the retaining walls made of reinforced concrete, it can be clearly seen that the reinforced soil intervention works obtained the highest scores from the sustainability point of view. This points out the fact that using geogrids for soil reinforcement is a much more efficient solution from an ecological perspective, following energy consumption and gas emissions, but also from an economic aspect, analyzing both the cost of the materials used and the labor, compared to the case of reinforced concrete as a construction material. Ensuring environmental sustainability is an admirable activity that civil engineers should definitely perform, not only in the preservation or renovation of existing structures but also in the design of future infrastructure. What engineers design and build today will have a long-term impact on the environment and society.

**Author Contributions:** Conceptualization, M.R.T.; Methodology, M.R.T. and C.I.B.; Validation, M.R.T.; Formal analysis, M.R.T.; Investigation, M.R.T.; Data curation, M.R.T.; Writing—original draft, M.R.T.; Writing—review & editing, M.R.T.; Supervision, M.R.T. and C.I.B. All authors have read and agreed to the published version of the manuscript.

**Funding:** This research received no external funding.

**Institutional Review Board Statement:** Not applicable.

**Informed Consent Statement:** Not applicable.

**Data Availability Statement:** Data are available from the first author. The literature data are in accordance with the references inserted.

**Conflicts of Interest:** The authors declare no conflicts of interest.

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
