# Peer review of "Intervention Works Conducted to Ensure the Stability of a Slope: A Sustainability Study"

_sustainability, doi:10.3390/su16041544_

Round 1

Reviewer 1 Report

Comments and Suggestions for Authors

Issue 1. This paper explores the sustainability aspects of interventions aimed at stabilizing a slope for an access road to an ecological landfill in Romania. The study assesses four different slope configurations using the Bob–Dencsak Specific Model, considering environmental, economic, and social parameters. A comparison of solutions reveals that the variant with reinforced soil, specifically using biaxial geogrid, achieves better energy efficiency and the highest sustainability index among the options studied.

Issue 2. Understanding the sustainability implications of slope stabilization interventions is crucial for making informed choices in construction projects, minimizing environmental impact, and promoting long-term ecological balance.

Issue 3. The Introduction section provides a comprehensive literature review. However, it lacks sufficient analysis of studies conducted in a broad context. This aspect needs significant improvement to provide a more balanced perspective. Please consider the suggested research (comes from, Indonesia, Algeria, Ukraine etc.).

Saptono, S., & Rezky, D.M. (2022). Sensitivity analysis of nickel haul road embankment slopes using the coefficient of variation approach. Min. Miner. Depos. 16(3), 48-53. Doi:10.33271/mining16.03.048

Boubazine, L., Boumazbeur, A., Hadji, R., & Fares, K. (2022). Slope failure characterization: A joint multi-geophysical and geotechnical analysis, case study of Babor Mountains range, NE Algeria. Min. Miner. Depos., 16(4), 65-70. Doi:10.33271/mining16.04.065

It will show that the investigated problem  is important not only in Romania.

Issue 4. In the comparison of solutions, what factors led to the reinforced soil with biaxial geogrid, featuring a slope of 1:1, being identified as the most sustainable option despite having a slightly lower safety factor?

Issue 5. How did the analysis account for the trade-off between safety factors and sustainability indices in choosing the optimal slope stabilization solution, particularly in the context of the minimum accepted safety factor of 1.50?

Issue 6. What were the key parameters studied in the analysis of the retaining wall made of reinforced concrete with a height of 6.40m, and how did its unfavorable results in terms of energy, costs, and sustainability index impact its ranking among the options?

Issue 7. In terms of energy consumption, how significant were the differences observed between interventions with reinforced concrete and those with reinforced soil, and how might these findings influence decision-making in future civil engineering projects?

Issue 8. The result section is too short. Moreover, there is a lack of Discussion. These sections must be enhanced.

Issue 9. Please provide a short description of further research (address e a short description of further research).

Issue 10. The novelty of the paper must be highlighted in the conclusions section (the authors should clearly and in detail indicate the fundamental scientific novelty of the paper and highlight it in the conclusions section)

Issue 11. In general, I must admit that a very good study was performed, and I will recommend your paper for publication after careful revision.

Author Response

Thank you very much for the feedback provided. As far as I managed, I tried to solve the problems you mentioned. Point by point I will respond to each request.

Reviewer 2 Report

Comments and Suggestions for Authors

This article is a good one and we have reviewed your paper carefully and would like to provide you with some feedback on our findings, the topic of this article is a long-term sustainability study of 4 intervention works to ensure slope stability. This includes the investigation of specific environmental, economic and social parameters. The main aim is to compare different solutions to determine which one is the most effective from a sustainability perspective.

But before publication can be considered, there are a number of issues that must be resolved. If the following problems are well solved, the reviewer believes that this paper has an important contribution to the study of slope stability.

1. The introduction should provide background, research gaps, and research objectives, but the author spends a lot of part explaining the background. It is recommended that you revise the details in the introduction to make it easier for the reader to understand the article better.

2. Abstract and keywords are too tedious, and there is no concise summary of the full text. It is recommended that you revise the abstract to more clearly summarize your findings.

3. Another obvious problem in this paper is the insufficient interpretation of the simulation results. It is recommended that you need to explain your simulation results in detail and why you got the results you did.

4. Your manuscript needs to be edited carefully, paying special attention to English grammar, spelling and sentence structure. Please review the manuscript so that the reader has a clear understanding of the objectives and results of the research.

Overall, your research work has some potential, but needs to be further refined and improved. I believe that through your careful consideration and appropriate modification of my suggestions, your paper will be further improved and promoted. Therefore, we recommend that you amend your manuscript accordingly and resubmit it for further consideration.

Author Response

(The authors gave the same response as above.)

Reviewer 3 Report

Comments and Suggestions for Authors

Dear [Author's Name],

I trust this message finds you well. I have carefully reviewed the manuscript submitted and would like to provide some constructive feedback for your consideration. Please note that my intention is to assist in enhancing the clarity and coherence of your work.

Abstract:

I suggest a restructuring of the abstract to ensure a more logical flow of ideas. Currently, it appears that various thoughts are grouped together, making the overall narrative less direct in defining the intentions. For instance, the section on the sustainability study could benefit from a more concise and ordered presentation. To illustrate, the opening statement could be refined for greater precision: "The aim of this paper is to evaluate utilising the Bob-Dencsak method a sustainable slope design considering environmental, economic and safety variable. It was collected 4 interventions by exploring on the ground…”

Introduction:

The narrative concerning energy and sustainability requires a clearer connection between these concepts. Perhaps exploring how a slope's efficiency in energy production relates to the broader theme of sustainability could strengthen the introduction. Additionally, considering an approach that emphasizes safety over energy consumption might be worth exploring for a more impactful introduction. Furthermore, when citing Shen et al.'s work, it would be beneficial to explicitly connect the technologies discussed to the slope's construction materials and design, thereby establishing a more evident link to sustainability.

Material and Methods:

  • The effectiveness of FEM in comparison with UK, Bob-Dencsak, and LEED could be better elucidated. Clarifying the connection between these different methodologies would enhance the reader's understanding.
  •  

3.2. The intervention works analyzed on the slope lack a clear link to the preceding questions, and the formulas presented may benefit from contextualization within the literature review. It would be helpful to explicitly state whether the evaluation pertains to the economic design or the safety design of the slope. Given that the focus is on sustainability, emphasizing how this approach differs from conventional safety-first designs would provide valuable insight for the reader.

What the contribution and novelty of this paper to another scholar who wants to research on sustainable design of slopes? a Confirmation case of the literature or a proof of concept? Would you be able to elaborate more on this paper.

Why should be remarkable to have a sustainable slope design?

I expect a discussion between methods and identify gaps in evaluating safety against sustainable designs.

Thanks

Comments on the Quality of English Language

The order of some sentences where not directly expressed as statements. There were more vague and in some cases the link between ideas may be lost for the reader. 

Author Response

(The authors gave the same response as above.)

Round 2

Reviewer 1 Report

Comments and Suggestions for Authors

I think that the authors have adequately addressed the comments made by the reviewers in the revised version of the manuscript. Therefore, I have no further comments. 

My congratulations to you for your diligent work.

All the best !!!

Reviewer 2 Report

Comments and Suggestions for Authors

The authors have modified the paper well.

Reviewer 3 Report

Comments and Suggestions for Authors

Dear Authors,

Thanks for your effort in improving the manuscript. Despite long sentences, the research produced a significant contribution or idea for further research: "Human safety in a given context may or may not ensure sustainability"

I wish you the best!